SOFTWARE

# PON-Del predictor for sequence retaining protein deletions

**Haoyang Zhang**, **Muhammad Kabir**, **Mauno Vihinen***

Department of Experimental Medical Science, Lund University, Lund, Sweden

* mauno.vihinen@med.lu.se

## Abstract

Protein deletions are frequent among both disease-causing and tolerated variants. Several mechanisms at the DNA, RNA and protein levels can lead to deletions. Many deletions are misclassified in the literature and databases, especially when the mRNA is degraded by the cellular quality-control mechanism. We developed a novel predictor for sequence retaining protein deletions, i.e., variants that do not alter the sequence downstream of the deletion site. We collected an extensive dataset of verified protein deletions, each described by a comprehensive set of context, content, position, and gene-based features. We evaluated both statistical and deep learning algorithms and selected a gradient boosting–based approach to develop the PON-Del predictor for short, 1–10 amino acid, sequence-retaining deletions. Variants are typically classified into two categories: either pathogenic or benign. However, there is always a third class of variants: variants of uncertain significance (VUSs), which have been ignored by all previous methods. PON-Del is the first deletion interpretation method that includes VUSs. It provides two outputs, binary and three-state prediction with VUSs. The performance of PON-Del was superior to that of previous methods. The tool is freely available at https://structure.bmc.lu.se/pon_del/.

## Author summary

Protein deletions are frequent among both disease-causing and tolerated variants, and are caused by several mechanisms at the DNA, RNA and protein levels. The reliable prediction of the effects of deletions is challenging. We developed a predictor for sequence retaining protein deletions, variants that do not alter the sequence beyond the deletion site. We collected an extensive dataset of verified protein deletions, and a comprehensive set of features to describe them. We evaluated seven algorithms and selected a gradient boosting–based approach to develop the PON-Del predictor for short, 1–10 amino acid, sequence-retaining deletions. Variants have typically been classified as pathogenic or benign. This practice misses the third category: variants of uncertain significance (VUSs). PON-Del is the first deletion interpretation method that

**Data availability statement:** The data used to train and test PON-Del is available in VariBench at https://structure.bmc.lu.se/VariBench/data/variationtype/deletions/Dataset2/data_pondel.csv and at the PON-Del website at http://structure.bmc.lu.se/pon-del. The code used to train PON-Del is available at https://github.com/zhanghaoyang0/pon_del_public.

**Funding:** The author(s) received no specific funding for this work.

**Competing interests:** The authors have declared that no competing interests exist.

includes VUSs. The performance of PON-Del was superior to that of previous methods. The tool is freely available at https://structure.bmc.lu.se/pon_del/.

## Introduction

Protein deletions are frequent variations. Among the close to 4.0 million variants in ClinVar [1], 5.0% are deletions (December 2025). Of these, 60% are (likely) pathogenic, and 19% are (likely) benign. Many protein deletions are tolerated, and natural length isoforms are common. Statistical analysis of natural and disease-related protein deletions revealed clear differences between the groups [2], for example, in the functions of deletion-containing proteins and in the sequence context of deletions. Further, pathogenic and benign variants differed, e.g., in size distribution, location relative to duplicated genes, domains, and protein termini, as well as the sequence context of deletions.

In the Variation Ontology [3], deletions are classified as sequence retaining or amphigoric. The former does not change the amino acid sequence after the deletion. The latter are due to mRNA frameshift variations and alter the C-terminal sequences of the encoded proteins. Many mRNA frameshift variations are misclassified in databases and literature at the protein level, when no protein is produced due to mRNA quality control mechanisms, such as nonsense-mediated decay (NMD) [4].

Depending on the structural location, deletions can be tolerated or harmful. N- and C-terminal truncations shorten the polypeptide chain from one end. The third category of deletions is internal deletion. Many mechanisms cause the deletions at the DNA, RNA, and protein levels. For a classification, see [5]. DNA deletions affect the coded protein unless the mRNA is degraded by NMD [6,7] or related mechanisms. The effect of NMD has been neglected in many articles and databases, leading to unreliable classification of deletions [4].

Deletions can also arise during transcription and translation. Alternative initiation and termination, along with alternative splicing, are common RNA changes that lead to deletions [8]. Mutually exclusive exons are a special case of alternative splicing [9]. Information for natural protein length variations can be obtained, e.g., from UniProtKB [10].

The effects of most RNA-level frameshift variations are obvious, as no protein is produced. For these, no prediction methods are needed. Sequence retaining deletions (in-frame variants at the RNA level) can be associated with disease or tolerated. Prediction methods have been developed for the disease relevance of deletions. The tools for sequence retaining deletions include CAPICE [11], DDIG-in [12], FATHMM-indel [13], INDELpred [14], KD4i [15], MutPred-Indel [16], PROVEAN [17], SHINE [18], and SIFT Indel [19]. VEST-Indel [20] is for both amphigoric and sequence-retaining deletions. The performance of the tools has been benchmarked with variants from gnomAD, ClinVar, and the DDD study [21]. The methods displayed a wide range of performances. The Matthews correlation coefficient (MCC) for the best methods was 0.68 [21], far from perfect.

These tools differ in both feature sets and algorithms. Various tree-based models have been the most popular, including decision trees (SIFT Indel), random forests (VEST-Indel), and gradient boosting methods (CAPICE, INDELpred). Other machine learning approaches have been support vector machines (DDIG-in and FATHMM-Indel), neural networks (MutPred-Indel), transfer learning (SHINE), and symbolic learning based on inductive logic programming (KD4i). PROVEAN relies on evolutionary conservation scores derived from sequence alignments. In addition to the algorithm and datasets, the methods differ regarding the features used for training.

We collected an extensive verified dataset of protein deletions and used it to train a predictor for short, 1–10 amino acid, sequence-retaining deletions. We evaluated both statistical and deep learning (DL) algorithms. Gradient boosting was chosen to train PON-Del. The performance was superior to that of previous methods. The method can be used to predict deletions in any protein by using a freely available web service. Variant effects are typically classified into two categories: either pathogenic or benign. Recently, we showed that all substitution variants of uncertain significance (VUSs) cannot be classified as pathogenic or benign, even with additional data and functional studies, due to natural biological heterogeneity [22]. Thus, VUSs must also be included in other types of variation interpretation tasks, including the classification of deletions. Therefore, PON-Del provides two outputs, binary and three-state prediction with VUSs.

## Design and implementation

The workflow of PON-Del was as follows (Fig 1). Briefly, data and features were collected from multiple sources. Deletions were obtained from ClinVar [1], LOVD [23], dbSNP [24], and UniProtKB [10]. Data cleaning was performed to remove duplicates. Deletions were mapped to the Matched Annotation from NCBI and EMBL-EBI (MANE)-selected transcripts and consist solely of short deletions. Duplicates or long deletions were removed. The data were then split into training and test sets to ensure a balanced representation of pathogenic and benign deletions. Feature selection was performed on the training data by removing low-frequency, zero-variance, and highly correlated features. Then, seven statistical and deep learning frameworks were evaluated using cross-validation to identify the best-performing model. In the end, the best framework was optimised for PON-Del through feature set refinement and hyperparameter tuning. To identify cases for which a binary classifier should abstain from prediction due to low confidence, we repeated the train–test split four additional times and retrained the model, resulting in 25 independently trained models (five folds across five splits). Prediction uncertainty was quantified using p-values derived from bootstrap resampling of the probability distributions and used to classify VUSs.

## Data collection and cleaning

We collected protein deletion variants from multiple sources to ensure comprehensive coverage. From the dbSNP [24], we obtained 265 variants by filtering for "inframe_deletion" variants with clinical significance (pathogenic, likely pathogenic, benign, or likely benign), limiting to deletion variation class and inframe deletion function class. From the LOVD database [23], we initially found 7,054 cases by selecting all transcript variants with clinical classification (benign or pathogenic), applying filters to exclude frameshift, termination, insertion, and ambiguous variants, and limiting to germline origin with valid genomic DNA change annotation. ClinVar [1] contained 2,256 deletion variants when searched for inframe_deletion variants with germline classification, clinical significance (pathogenic, likely pathogenic, benign, or likely benign), and deletion variation type. Additional benign deletions were collected from UniProtKB [10]. They represent natural protein isoforms and are classified as benign phenotypically. All human protein isoforms were collected from UniProtKB and matched with protein sequences corresponding to the MANE version 1.4 transcripts [25]. Pairwise sequence alignments were obtained using the needle algorithm for dynamic programming from the EMBOSS package [26]. After comparison to the MANE-based sequences, we identified 7,581 unique deletions from the UniProtKB.

The variants were mapped to MANE-select transcripts. The genomic positions of the variants were defined using TransVar [27]. The genomic deletion length was ≤ 30 nucleotides to avoid exon skip events, as only a few longer variants were identified. The protein deletion length was thus ≤10 amino acids. Protein variants from dbSNP and LOVD not

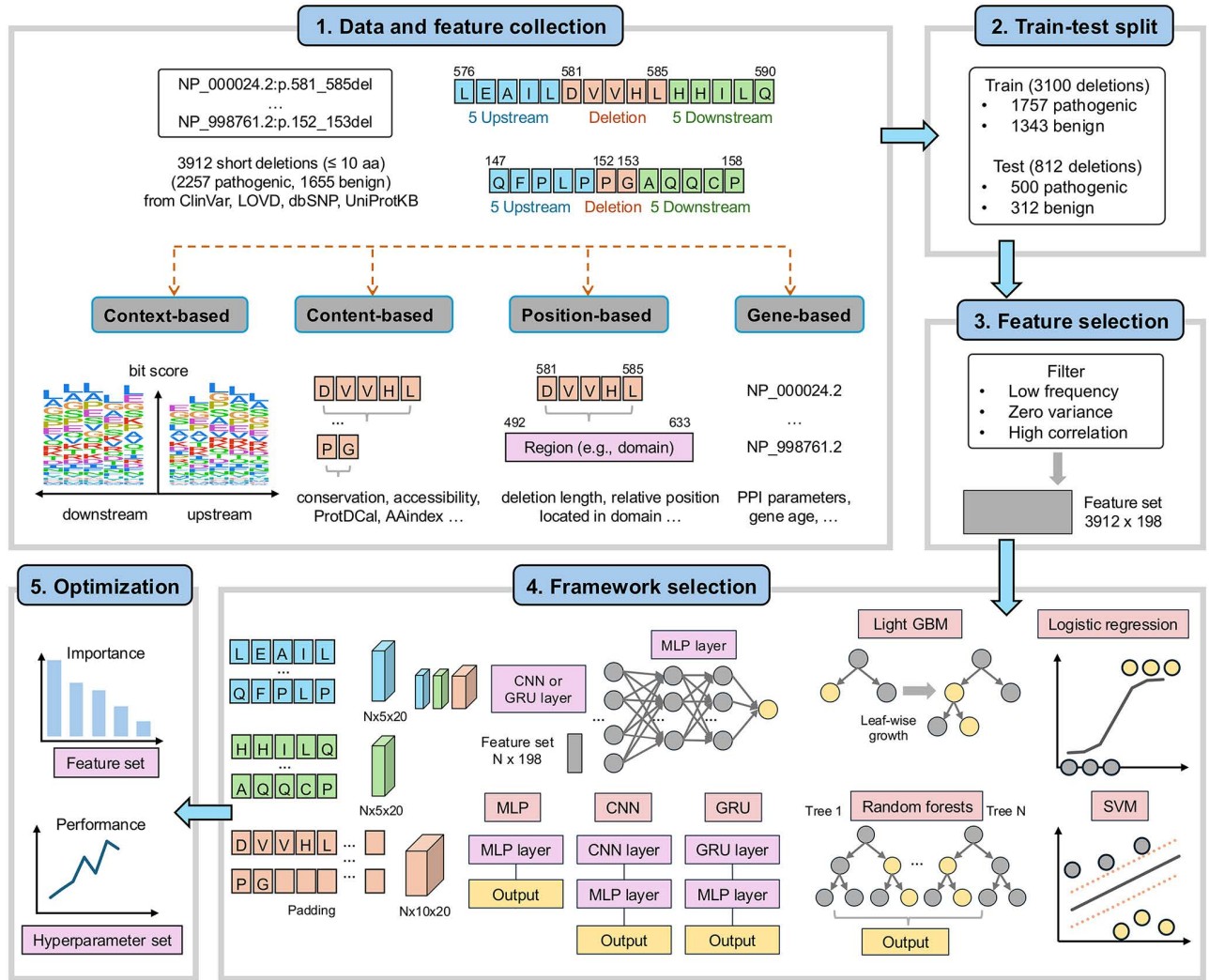

**Fig 1. Overview of the PON-Del development pipeline includes (1) data and feature collection.** We obtained 611 features in four categories. (2) The data were split into training and testing sets. (3) Feature selection involved removing non-informative, zero-variance, and highly correlated features. (4) Seven frameworks were tested to identify the best-performing one. (5) The selected model was optimised through feature refinement and hyperparameter tuning to develop the final PON-Del predictor.

present in MANE-select transcripts were annotated using VEP [28]. We then integrated ClinVar and UniProtKB protein deletions, mapping them to MANE-select transcripts, and removed duplicates both within and across data sources. Finally, we added UniProt identifiers to ensure proper protein identification. After processing, the final dataset comprised 4,243 deletions from MANE-select transcripts: 1,555 from ClinVar, 178 from dbSNP, 1,368 from LOVD, and 1,142 from UniProtKB. The collected deletion datasets are freely available in VariBench (Nair and [29]) https://structure.bmc.lu.se/VariBench/data/variationtype/deletions/Dataset2/data_pondel.csv.

## Feature collection

We obtained an extensive set of 611 features in four categories: context-based (N = 5), content-based (N = 561), position-based (N = 32), and gene/protein-based features (N = 13).

Context-based features described the sequence information within and surrounding deletion sites. We analysed five residues upstream and downstream of each deletion, calculating position-specific bit scores from sequence alignments of all the benign and pathogenic deletions to capture conservation patterns and amino acid preferences. Sequence logos obtained with the SeqLogo program [30] were used to analyse these patterns and to define bit scores for each amino acid in each position. Bit scores for segments before and after each deletion were summed and used as features. In addition to bit scores, we extracted three sequence segments (deletion, upstream, and downstream), each padded or trimmed to 5 residues to ensure fixed-length input. These features were used specifically in the DL models.

Content-based features described the average properties of the deleted regions. Conservation was assessed using a position-specific scoring matrix (PSSM), and we used the information content per position (PSSM 3) [31] to assess the degree of conservation at each position. The pathogenicity score for each deletion was calculated as the average of the PON-P3 pathogenicity scores of all possible substitutions within the deleted region [31].

As previously described [31], the structures of MANE-selected proteins were predicted and used to obtain the solvent-accessible surface areas (SASA) of the original residues using FreeSASA [32]. Averages for deleted residues were calculated and used as a feature. The average amino acid physicochemical properties in the deletions were calculated with AAindex (553 features) [33] or ProtDCal propensities (16 features) [34], as previously described [31].

Position-based features described the localisation of deletions within the protein. The genomic start and end positions were annotated with start and end positions in protein using TransVar [27]. We calculated deletion length and relative position, along with the closest distance to the protein terminus.

We annotated whether deletions overlapped with functional regions, including domains, palindromes, repeat regions, transmembrane (TM) regions, intrinsically disordered regions (IDRs), the last exon within a gene (with a 50 bp window), and 16 functional regions from SwissProt. Domains were identified with InterPro [35], repeats with T-REKS [36], and IDRs based on data in DisProt [37]. The Human Transmembrane Proteome database [38] provides information for TM. Palindromic regions were determined by identifying substrings within protein sequences using a comparison between each substring and its reverse with the R package "seqinr". The last genomic exon regions were obtained from the MANE 1.4 GFF file. The secondary structural assignments were determined from protein structures or models obtained with Alpha-Fold [39; 40] with STRIDE [41] and included eight classes: α-helix, β-strand, B-beta, b-beta, turn/coil, G-helix, π-helix, and low-confidence regions from AlphaFold models [40].

The gene/protein-based features describe gene-level characteristics. Classifications included housekeeping, haplo-insufficient, and redundant genes, pseudogenes, or duplicated genes, see PON-P3 [31]. Six protein-protein interaction (PPI) metrics (degree, closeness, betweenness, harmonic centrality, hub score, and power centrality) were computed with igraph [42] from the data in the STRING database [43] to indicate the network impact of proteins. The age of each gene was determined using ProteinHistorian [44]. For further details on the features, see [31].

After feature collection, we obtained a dataset of 3,912 unique deletions with complete feature information. The dataset was then split into a training set (3,100 deletions: 1,757 pathogenic and 1,343 benign) and a test set (812 deletions: 500 pathogenic and 312 benign) for cross-validation and final evaluation, see Table 1. Four additional train–test splits were generated to assess model robustness and to compute bootstrap p-values for VUS classification.

## Feature filtering

To refine the feature set, we selected features using the training data. For binary features (such as position-based features and gene/protein classifications), we removed columns with a minority class of 5 or fewer samples, as these features were considered unstable.

For numerical features, we first removed those with zero variance, as they had no discriminative power. Next, we computed the Spearman correlation matrix for the remaining features. For any pair of features with an absolute Spearman

**Table 1. Number of variations and proteins in the training and test datasets.**

| Train-test split[a] | CV training | | Blind test | | Total | |
|---|---|---|---|---|---|---|
| | Pathogenic[b] | Benign | Pathogenic | Benign | Pathogenic | Benign |
| v1 | 1757 (56.7) | 1343 (43.3) | 500 (61.6) | 312 (38.4) | 2257 | 1655 |
| v2 | 1841 (58.0) | 1333 (42.0) | 416 (56.4) | 322 (43.6) | | |
| v3 | 1771 (57.0) | 1337 (43.0) | 486 (60.5) | 318 (39.5) | | |
| v4 | 1832 (58.0) | 1326 (42.0) | 425 (56.4) | 329 (43.6) | | |
| v5 | 1765 (56.8) | 1342 (43.2) | 492 (61.1) | 313 (38.9) | | |

[a]The train–test splits used stratified sampling by protein-level classification (pathogenic vs benign), to ensure similar pathogenic to benign proportions and that no protein appeared in both sets.

[b]Numbers in brackets are percentages.

correlation greater than 0.8, one feature was removed to reduce redundancy. The three sequence-based inputs (deletion, upstream, and downstream segments) were not filtered. They were used in the DL models.

After these steps, the number of features was reduced from 611 to 195.

## Framework selection

We compared altogether seven algorithms, four statistical learning methods, and three DL approaches, using 5-fold cross-validation.

The statistical learning methods included the gradient boosting algorithm LightGBM [45], logistic regression (LR), random forests (RF) [46], and support vector machine (SVM) [47]. LR was applied with balanced class weights and MinMax-scaled features. RF was constructed with 100 estimators to ensure robust ensemble predictions. SVM was implemented with a linear kernel and probability estimation enabled.

For the deep learning approaches, we implemented three architectures: a multi-layer perceptron (MLP) [48], convolutional neural network (CNN) [49], and gated recurrent unit (GRU) [50]. The MLP used a three-layer architecture with layer normalisation and LeakyReLU activation. The CNN combined sequence processing with feature analysis through a hybrid architecture. It processed three sequences (deletion, upstream, and downstream) in parallel, concatenated the extracted features, scaled them by a factor of 0.2, and fed them into a fully connected branch with layer normalisation and LeakyReLU activation functions. The GRU employed a bidirectional design to process sequence data, with both sequence and non-sequence features processed in parallel branches. Similar to the CNN, the GRU processed three sequences in parallel, concatenated and scaled the features by a factor of 0.2, and fed them into a fully connected branch.

All the DL models were trained using the Adam optimiser with a learning rate of 0.001 and binary cross-entropy loss. For models not using tree-based methods, features were scaled using MinMax normalisation. Early stopping was implemented with a patience of 30 epochs, based on validation loss. The final models were trained on the complete training dataset using the optimal hyperparameters and feature sets identified during the optimisation process.

## Model optimisation

To optimise the performance of the selected framework, we applied model-specific optimisation strategies. As the best framework was a tree model, recursive feature extraction (RFE) [51] was used to assess feature importance by evaluating feature sets ranging from the top 10 to the top 190 features, with an interval of 10 features between each set. This approach allowed us to identify the optimal number of features. Then, we employed an extensive hyperparameter optimisation framework Optuna 4.2.1 [52], which automates the search for the best parameters. The optimisation process

involved defining a search space for key hyperparameters over 100 trials. The best parameters identified through this process were then used to train the final PON-Del model.

## Predictions and classification schemes

PON-Del provides two types of predictions. A traditional binary predictor outputs scores between 0 and 1, with values above 0.5 indicating pathogenicity and those below 0.5 indicating benignity. We defined VUSs based on the consistency of PON-Del predictions across multiple models. For each variant, we collected 25 predicted probabilities from independent PON-Del models and used bootstrap resampling (1,000 iterations) to estimate the sampling distribution of the mean predicted probability. We then performed a two-sided hypothesis test against a null mean of 0.5 (no evidence for pathogenicity versus benignity) and computed a p-value from the bootstrap distribution.

Variants whose mean predicted probability was significantly different from 0.5 ($p < 0.05$) were assigned to pathogenic or benign according to whether the predicted probability was $> 0.5$ or $\leq 0.5$, respectively. Variants for which we could not reject the null hypothesis ($p \geq 0.05$) were considered VUS, reflecting insufficient statistical evidence to confidently assign them to either the pathogenic or benign class.

## Performance evaluation

A systematic performance assessment was performed according to the published recommendations [53, 29]. The measures included positive predictive value (PPV), negative predictive value (NPV), sensitivity, specificity, accuracy, the MCC, the overall performance measure (OPM) [54], and area under the curve (AUC) as follows

$$PPV = \frac{TP}{TP + FP}$$

$$NPV = \frac{TN}{TN + FN}$$

$$Sensitivity = \frac{TP}{TP + FN}$$

$$Specificity = \frac{TN}{TN + FP}$$

$$Accuracy = \frac{TP + TN}{TP + TN + FP + FN}$$

$$MCC = \frac{(TP \times TN) - (FP \times FN)}{\sqrt{(TP + FN) \times (TP + FP) \times (TN + FN) \times (TN + FP)}}$$

$$OPM = \frac{(PPV + NPV)\,(Sensitivity + Specificity)(Accuracy + (1 + MCC)/2)}{8}$$

TP and TN are correctly predicted pathogenic and neutral cases, respectively, and FN and FP are the numbers of incorrect predictions for pathogenic and neutral cases, respectively.

To address the slight class imbalance between pathogenic and benign variants, normalised metrics were calculated by adjusting the number of pathogenic variants to match the number of benign ones. A classification threshold of 0.5 was

used for threshold-dependent metrics to distinguish between pathogenic and benign deletions. In contrast, AUC provides a threshold-independent evaluation of model performance.

## Results

Both benign and pathogenic, verified sequence retaining deletions were collected from ClinVar [1], dbSNP [24], LOVD [23], and UniProtKB [10]. The clinical significance (pathogenic, benign) of the variants was retrieved from ClinVar, dbSNP, or LOVD. All these variants are sequence retaining and thereby produced. Further benign variants included natural protein isoforms from UniProtKB.

We collected a total of 611 features for the deletions. After data cleaning and feature collection, there were in total 3,912 unique deletions with a full set of features (Table 1). The deletions appear in several different proteins; the total number of proteins was 1904. The distribution of all deletion lengths is shown in the S1A Fig and S1C Fig in for training and test sets. Far majority of both pathogenic and benign deletions were of one residue long, the distribution is shown in S1B Fig. Given that most deletions were short and few were longer than 10 amino acids, we decided to train a predictor for deletions of 1–10 amino acids. The developed tool might work even with longer deletions, but since there were not enough cases for training and testing, the range of deletions was limited.

The variants were divided into training (N = 3100) and test (N = 812) data sets, both of which included a wide variety of proteins (Table 1, v1). The two data sets do not share any proteins. In the training set, benign and pathogenic variants had mean deletion lengths of 4.11 (SD = 2.87) and 2.71 (SD = 2.26), respectively. In the test set, the means were 4.51 (SD = 2.86) for benign and 2.51 (SD = 2.14) for pathogenic variants. The length differences were very small.

The flowchart of the method development is depicted in Fig 1. We collected an extensive set of features, tested seven algorithms, and hyperparameter-tuned the best-performing one.

### Feature selection and choice of algorithm

We started with an extensive set of features. Initially, there were 611 features. The features were grouped into four categories. Context features describe the sequence environment of deletions. Content-based features are properties averaged over the deleted region and include, e.g., sequence conservation and physicochemical propensities. Position features capture the relative location within the sequence and proximity to structural or functional regions. Gene/protein features include PPI parameters, gene age, and localisation to functional regions.

First, we investigated which features were informative. The variances for three features were close to 0, so they were excluded. Two binary features with fewer than 5 annotated instances were removed. Next, we defined the Spearman correlation for all feature pairs. This led to the removal of 408 features with high (>0.8) correlations. With the stepwise process, we reduced the number to 195.

We used the remaining 195 features to train seven methods, i.e., CNN, GRU, LightGBM, LR, MLP, RF, and SVM. The results with 5-fold CV are shown in Table 2. Logistic regression, one of the simplest algorithms, was used to define the baseline performance. LightGBM achieved the best overall performance across most metrics, with an AUC of 0.91, accuracy of 0.83, MCC of 0.66, and OPM of 0.58 (Table 2). RF was the next best-performing method, followed by the DL models.

The set of 198 remaining features was used for final feature selection by training LightGBM predictors with feature counts ranging from 10 to 190, in increments of 10. For results, see S2 Fig. To avoid the so-called curse of dimensionality — excessive numbers of features— we chose the smallest number of features that provided optimal performance. RFE was used to select the most important features. The performance was as good with the 20 features as with versions with larger numbers of features (S2A Fig). Therefore, we chose 20 features for training the final predictor, called PON-Del. Deletions are not predicted in the first position since the removal of the start codon would prevent protein synthesis.

Next, we optimised the predictor by tuning hyperparameters over 100 trials (S2B Fig). The tested parameters were, in order of significance, learning rate, boosting type, number of leaves, reg alpha, bagging fraction, reg lambda, minimum

**Table 2. Comparison of the performance of different algorithms in 5-fold CV. The best-performing method is shown in bold.**

| Metrics | LightGBM | RF | LR | SVM | MLP | CNN | GRU |
|---|---|---|---|---|---|---|---|
| AUC | **0.91** | 0.88 | 0.88 | 0.87 | 0.89 | 0.89 | 0.88 |
| OPM | 0.58 (**0.58**) | 0.54 (0.53) | 0.5 (0.5) | 0.5 (0.49) | 0.53 (0.53) | 0.53 (0.53) | 0.54 (0.54) |
| Accuracy | 0.84 (**0.83**) | 0.82 (0.81) | 0.79 (0.79) | 0.8 (0.79) | 0.81 (0.81) | 0.81 (0.81) | 0.82 (0.81) |
| MCC | 0.66 (**0.66**) | 0.62 (0.62) | 0.59 (0.59) | 0.58 (0.58) | 0.62 (0.62) | 0.62 (0.62) | 0.63 (0.63) |
| PPV | 0.85 (**0.81**) | 0.82 (0.77) | 0.83 (0.79) | 0.81 (0.76) | 0.84 (0.8) | 0.82 (0.78) | 0.85 (**0.81**) |
| NPV | 0.82 (**0.86**) | 0.82 (**0.86**) | 0.75 (0.8) | 0.78 (0.82) | 0.78 (0.82) | 0.81 (0.84) | 0.77 (0.81) |
| Sensitivity | 0.86 (0.86) | 0.88 (**0.88**) | 0.79 (0.79) | 0.84 (0.84) | 0.83 (0.83) | 0.86 (0.86) | 0.81 (0.81) |
| Specificity | 0.8 (0.8) | 0.74 (0.74) | 0.79 (0.79) | 0.73 (0.73) | 0.79 (0.79) | 0.75 (0.76) | 0.82 (**0.82**) |
| TP | 303.6 (303.6) | 307.6 (307.6) | 279.2 (279.2) | 295.8 (295.8) | 290.4 (290.4) | 301.6 (301.6) | 285.8 (285.8) |
| TN | 214.0 (280.2) | 198.4 (259.6) | 212.8 (278.2) | 197.0 (257.8) | 213.8 (279.6) | 202.8 (265.4) | 219.0 (286.4) |
| FP | 54.6 (71.2) | 70.2 (91.8) | 55.8 (73.2) | 71.6 (93.6) | 54.8 (71.8) | 65.8 (86.0) | 49.6 (65.0) |
| FN | 47.8 (47.8) | 43.8 (43.8) | 72.2 (72.2) | 55.6 (55.6) | 61.0 (61.0) | 49.8 (49.8) | 65.6 (65.6) |

[a]The value inside the brackets is the normalised value.

child weight, bagging frequency, minimum split gain, feature fraction, and minimum number of child samples. Results in Table 3 show that the effect of optimisation was marginal. The performance was increased by less than 1%. The unnormalised results pertain to the ratio of pathogenic to benign variants in the dataset (58% vs 42%), whereas the normalised results pertain to balanced data.

Table 3 shows the results for all the tested methods on the blind test data. The best performance measures were scattered among the algorithms. PON-Del achieved the highest overall performance, with an AUC of 0.91, an accuracy of 0.82, an MCC of 0.65, and an OPM of 0.56. It also showed the high sensitivity (0.84) and strong precision (PPV: 0.81, NPV: 0.84) with balanced specificity (0.80). These results suggest that while different modelling frameworks exhibit varying performance, the choice of features has a greater impact on model effectiveness than the specific algorithm used. On the blind test set, GRU had the second-best performance. To evaluate the robustness of PON-Del predictions, we assessed performance under five independent train–test partitions. As shown in S3 Fig, PON-Del showed consistently high and stable performance over the splits and with very small differences between the splits.

We assessed how the p-value threshold used to define VUSs influenced the predictive performance of PON-Del. The evaluated thresholds ranged from 1.0 (equivalent to no uncertainty) down to 0.001. The performance increased together with decreasing threshold (S4 Fig). We did not determine an optimal p-value threshold, as stricter thresholds improve performance at the cost of increasing the number of unclassified (missing) predictions. As a practical solution, we used 0.05 as the threshold.

## Interpretability analysis

To gain insight into the reasoning of the predictor, we investigated the selected features. Since most ML methods are largely black boxes, their interpretability is a concern. Of the tested algorithms, the decision process is intuitively understandable only for logistic regression. We used two analyses to assess the importance of the selected features and two analyses to investigate if the datasets were biased.

The Shapley plot in Fig 2A shows the importance of the features to pathogenicity (positive values) and benign (negative values) prediction. Colour indicates the range of feature values; blue indicates low values, and red indicates high values. In the binary features, a missing property is indicated by a zero (blue), and the presence of the property is indicated by a red value. For features with a range of values, the colour scale indicates the increasing feature value.

**Table 3. Comparison of the performance of different algorithms on the blind test set. The best-performing method is shown in bold.**

| Metrics | PON-Del | RF | LR | SVM | MLP | CNN | GRU |
|---|---|---|---|---|---|---|---|
| AUC | **0.91** | 0.89 | 0.88 | 0.88 | 0.89 | 0.89 | 0.9 |
| OPM | 0.56 (**0.56**) | 0.53 (0.52) | 0.48 (0.49) | 0.48 (0.47) | 0.53 (0.53) | 0.53 (0.53) | 0.55 (**0.56**) |
| Accuracy | 0.83 (**0.82**) | 0.82 (0.8) | 0.79 (0.79) | 0.79 (0.78) | 0.82 (0.81) | 0.82 (0.81) | 0.82 (**0.82**) |
| MCC | 0.64 (**0.65**) | 0.61 (0.61) | 0.57 (0.58) | 0.56 (0.56) | 0.61 (0.62) | 0.61 (0.62) | 0.64 (0.64) |
| PPV | 0.87 (0.81) | 0.84 (0.77) | 0.86 (0.79) | 0.83 (0.75) | 0.86 (0.79) | 0.86 (0.79) | 0.88 (**0.82**) |
| NPV | 0.76 (**0.84**) | 0.77 (**0.84**) | 0.69 (0.78) | 0.73 (0.81) | 0.75 (0.83) | 0.75 (0.83) | 0.75 (0.83) |
| Sensitivity | 0.84 (0.84) | 0.86 (**0.86**) | 0.78 (0.78) | 0.83 (0.83) | 0.84 (0.84) | 0.84 (0.84) | 0.83 (0.83) |
| Specificity | 0.8 (0.8) | 0.75 (0.75) | 0.79 (0.79) | 0.72 (0.72) | 0.77 (0.77) | 0.77 (0.77) | 0.81 (**0.81**) |
| TP | 422 (422) | 430 (430) | 391 (391) | 417 (417) | 421 (421) | 421 (421) | 415 (415) |
| TN | 251 (402) | 233 (373) | 248 (397) | 225 (361) | 241 (386) | 241 (386) | 254 (407) |
| FP | 61 (98) | 79 (127) | 64 (103) | 87 (139) | 71 (114) | 71 (114) | 58 (93) |
| FN | 78 (78) | 70 (70) | 109 (109) | 83 (83) | 79 (79) | 79 (79) | 85 (85) |

[a]The value inside the brackets is the normalized value.

The selected 20 features were arranged in descending importance in the Shapley plot [55]. The most important feature is haploinsufficiency, followed by structural and functional features, including low-confidence secondary structural assignments, sequence conservation, accessibility, and location within a domain (Fig 2A).

Almost half of the features, nine, are for AAindex parameters of protein physical and chemical propensities. They included LEVM760103, side chain angle theta (AAR) [56]; RACS820103, average relative fractional occurrence in AL(i) [57]; GEOR030105, linker propensity [58]; SUEM840102, helix-coil stability constant [58]; ARGP820102, signal sequence helical potential [59]; NAKH920103, membrane protein amino acid composition [60]; CHOP780215, the frequency of the 4th residue in turn [61]; NAKH900104, membrane protein amino acid composition [60]; and BIGC670101, amino acid hydropathy [62]

Other types of features include protein length, closeness and hub score in the PPI networks, location within turns or coils, repeats, and in palindromes. Closeness and hubscores are overall measures of the topology of the PPI network. Low-confidence regions and turns and coils were the only protein secondary structural classes selected. Repeats and palindromes are short sequence stretches which either appear several times or read the same way from both ends.

As another analysis to highlight the decision-making process in PON-Del, we investigated the distribution of the values in the 20 features (Fig 2B). Haploinsufficiency and low-confidence AlphaFold predictions in the deleted region were the most important features. There are significant differences in the distribution patterns for pathogenic and benign deletions. Distinct patterns are also evident, e.g., in location to repeats, palindromes, turns or coils, domain localisation, sequence conservation, accessibility, and PPI scores. Further, physicochemical AAindex features show different distributions; however, the differences in these are smaller.

To investigate the impact of deletion origin on predictor performance, we evaluated the classifier separately on subsets restricted to each source individually, or combined with UniProtKB, when benign cases were insufficient in numbers (Clin-Var + UniProtKB, dbSNP + UniProtKB, LOVD + UniProtKB). Performance varied across databases (S5 Fig). The LOVD-only dataset differed from the others and showed the lowest performance for AUC, PPV, accuracy and MCC. This dataset contains only a very small number of benign deletions, which apparently are not fully representative. In summary, the combination of the datasets is the best option and was used to train and test PON-Del.

Next, we studied the performance for deletions in different types of proteins and genes, as well as in genes duplicated during human evolution (Fig 3). The functional features investigated included housekeeping, essential, and haploinsufficiency-related proteins. In all these categories, correct predictions were more common than misclassifications.

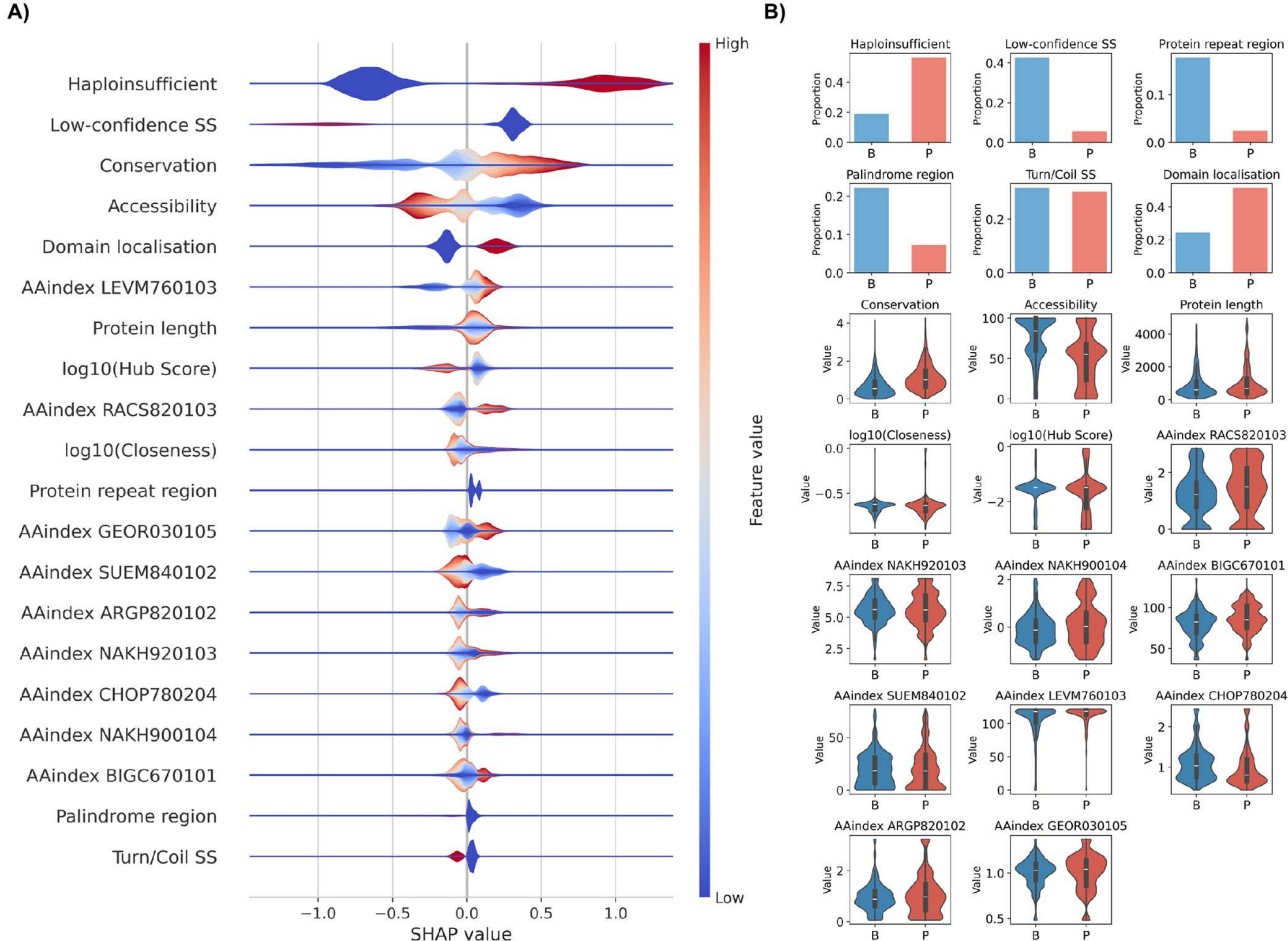

**Fig 2. Shapley plot and distribution of the scores for the 20 selected features, organised in descending order of importance. A)** The features are colored based on their value, ranging from blue to red. The SHAP value indicates the impact for both positive (pathogenic) and negative (benign) predictions. **B)** Distributions of the values for the selected features among pathogenic and benign data in the training set.

In pseudogenes, the performance was almost equal. In proteins for duplicated genes, correct predictions were somewhat more common. The proportions in Fig 3 indicate the occurrence of the features in the dataset.

Combined, the analysis provides a clear understanding of the features and their contributions to PON-Del predictions. Deletions affect sequences and structures in many ways, as evident from the wide spectrum of features. Three out of the four feature categories are represented among the selected features.

## Comparison to other tools

Several methods have been deployed to predict the pathogenicity of sequence retaining deletions. Some methods were excluded due to unavailability or outdated scripts that could not be installed. SIFT-Indel was excluded because it predicted just a tiny fraction of the test cases. We could not run DDIG-IN or KD4i because no code was available. CAPICE did not provide all the results. PROVEAN is no longer supported. SHINE is limited only to single-residue deletions. We used their pre-calculated scores for comparison.

We compared the performance of PON-Del to CADD, FATHMM-indel, INDELpred, MutPred-Indel, SHINE, and VEST-Indel (Table 4). These methods cover a wide range of different algorithms and approaches. We used the prediction

PLOS Computational Biology

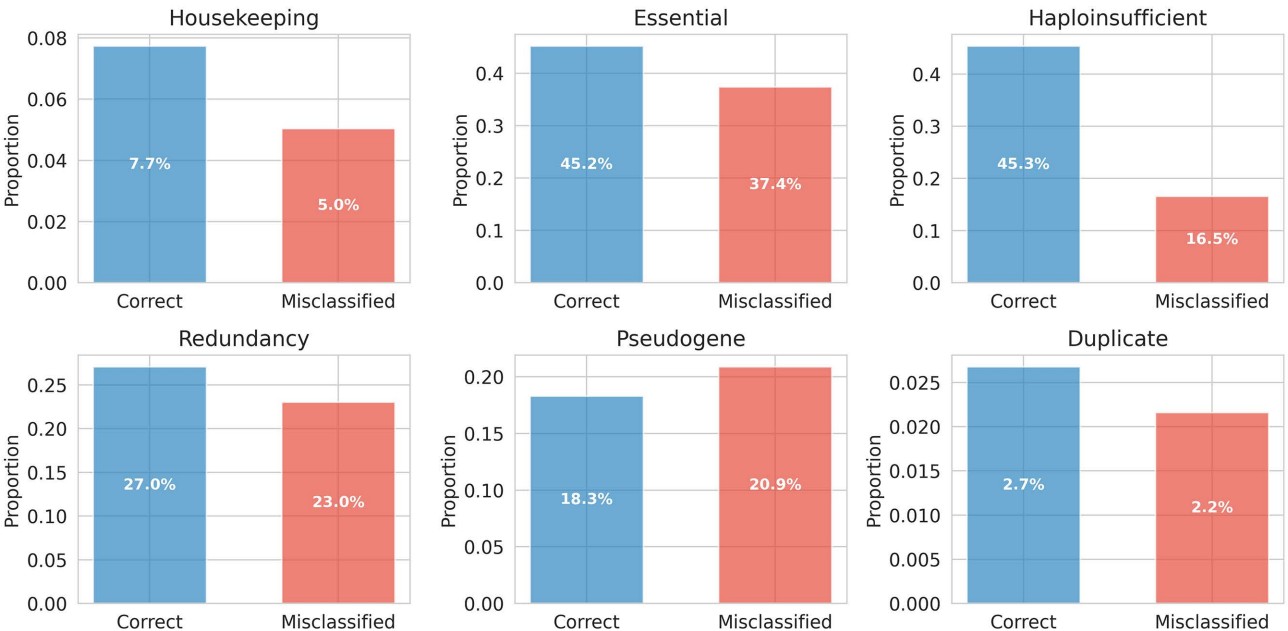

**Fig 3. Comparison of correct and incorrect deletion predictions in protein functional categories, pseudogenes, redundant proteins, and those originating from gene duplications.** The percentages are for the proportions of correctly predicted and misclassified variants for each feature. None of the categories classifies all the variants.

score to measure AUC, except for VEST-Indel. It provides only the p-value, which we used to calculate AUC. INDEL-pred and MutPred-Indel used as a threshold 0.5, and VEST-Indel used 0.05, based on its p-value. FATHMM-indel provided binary pathogenicity labels directly. For CADD, the pathogenicity threshold was set to 20, as recommended by the developers.

Table 4 shows that PON-Del 2-state and 3-state (with VUSs) versions outperformed the other deletion pathogenicity predictors on the blind test set, achieving the highest overall metrics. PON-Del 3-state predictor obtained the highest scores for AUC (0.92), accuracy (0.83), MCC (0.66), and OPM (0.57). It also demonstrated balanced performance with high PPV (0.8), NPV (0.87), sensitivity (0.88), and specificity (0.78). The 2-state predictor had the best PPV of 0.81 and specificity of 0.8. While FATHMM-indel had the highest sensitivity (0.97) and NPV (0.83), its poor PPV (0.56) and specificity (0.24) reflect a strong positive bias. The other tools showed lower and less balanced performance, while VEST-Indel performed relatively well (AUC: 0.88, accuracy: 0.82). These results highlight that the superior performance of PON-Del is primarily driven by its carefully engineered features.

The other methods have been trained on ClinVar data; therefore, their performance measures are likely inflated by circularity. Thus, their true performance is even lower than the benchmark showed.

Table 4 contains results for the two versions of PON-Del, the two-state pathogenic-benign predictor and the three-state predictor including VUSs. The two-state prediction is an oversimplification, since in reality, there are also always VUSs that cannot be classified along the binary pathogenic-benign axis [22]. The results in Table 4 show that the three-state predictor achieved almost identical performance to the binary tool. Because three-state prediction is more difficult and VUSs can overlap with pathogenic or benign variants, the result indicates that accommodating uncertainty did not compromise predictive power.

We could not test the performance with verified VUSs due to the lack of such data. If VUSs were included in the performance assessments, it would be apparent that the three-state predictor would outperform other tools. All other tools ignore VUSs; thus, in real-life situations, their performance is lower than shown in Table 4.

**Table 4. Performance comparison of deletion predictors on the blind test set. The best-performing method is shown in bold.**

| Metrics | PON-Del 2-state | PON-Del 3-state | CADD | FATHMM-indel | INDELpred | MutPred-Indel | SHINE | VESTIndel |
|---|---|---|---|---|---|---|---|---|
| AUC | 0.91 | **0.92** | 0.54 | 0.74 | 0.61 | 0.81 | 0.83 | 0.88 |
| OPM | 0.56 (0.56) | 0.58 (**0.57**) | 0.14 (0.14) | 0.31 (0.28) | 0.32 (0.32) | 0.38 (0.39) | 0.41 (0.45) | 0.55 (0.55) |
| Accuracy | 0.83 (0.82) | 0.84 (**0.83**) | 0.49 (0.52) | 0.69 (0.61) | 0.68 (0.69) | 0.73 (0.73) | 0.79 (0.76) | 0.83 (0.82) |
| MCC | 0.64 (0.65) | 0.66 (**0.66**) | 0.04 (0.04) | 0.33 (0.31) | 0.36 (0.37) | 0.45 (0.46) | 0.46 (0.53) | 0.63 (0.64) |
| PPV | 0.87 (**0.81**) | 0.87 (0.8) | 0.64 (0.53) | 0.67 (0.56) | 0.78 (0.69) | 0.82 (0.73) | 0.92 (0.74) | 0.87 (0.81) |
| NPV | 0.76 (0.84) | 0.8 (0.87) | 0.4 (0.52) | 0.83 (**0.88**) | 0.57 (0.68) | 0.62 (0.73) | 0.49 (0.79) | 0.76 (0.83) |
| Sensitivity | 0.84 (0.84) | 0.88 (0.88) | 0.39 (0.39) | 0.97 (**0.97**) | 0.68 (0.68) | 0.72 (0.72) | 0.81 (0.81) | 0.84 (0.84) |
| Specificity | 0.8 (**0.8**) | 0.78 (0.78) | 0.65 (0.65) | 0.24 (0.24) | 0.7 (0.7) | 0.74 (0.74) | 0.71 (0.71) | 0.8 (**0.8**) |
| TP | 422.0 (422.0) | 423.0 (423.0) | 196.0 (196.0) | 484.0 (484.0) | 338.0 (338.0) | 362.0 (362.0) | 215.0 (215.0) | 420.0 (420.0) |
| TN | 251.0 (402.0) | 231.0 (374.0) | 202.0 (324.0) | 76.0 (122.0) | 217.0 (348.0) | 230.0 (369.0) | 47.0 (188.0) | 245.0 (398.0) |
| FP | 61.0 (98.0) | 66.0 (107.0) | 110.0 (176.0) | 235.0 (378.0) | 95.0 (152.0) | 82.0 (131.0) | 19.0 (76.0) | 62.0 (101.0) |
| FN | 78.0 (78.0) | 58.0 (58.0) | 304.0 (304.0) | 16.0 (16.0) | 162.0 (162.0) | 138.0 (138.0) | 49.0 (49.0) | 79.0 (79.0) |
| N missing (%) | 0 (0) | 34 (0.42) | 0 (0) | 1 (0.1) | 0 (0) | 0 (0) | 482 (59.4) | 6 (0.7) |

[a]The value inside the brackets is the normalised value.

To evaluate the robustness of the methods, we assessed performance under five independent train–test partitions in method comparison (S5 Fig). The results are consistent over the partitions, indicating that the partitions are representative. The blind test set can thus be considered representative. PON-Del was consistently the best-performing tool, or among the best methods. The other methods showed more differences for the individual measures and less balanced predictions.

The results in S6 Fig indicate the consistency of different train-test splits. The performances are comparable to those in Table 4, and the differences in the splits are largely consistent for the tools, indicating that the datasets do not introduce major bias. The order for performances is the same as in Table 4. The results between the splits are the most consistent, i.e., most similar, for PON-Del.

## Test case

As an example of the method application, we show the distribution of the predicted single amino acid deletions in the Bruton tyrosine kinase, BTK. Among more than 500 human protein kinases, BTK contains the largest number of different disease-causing variants. There are several verified small deletions known and distributed in BTKbase, the database for BTK variants [63]. The positions of the disease-causing deletions and PON-Del predicted deletions are indicated in Fig 4. All the known short BTK-related deletions were predicted to be pathogenic. The positions of the harmful variations are shown in the protein three-dimensional structure in Fig 4B. They are distributed along the protein chain.

BTK is sensitive to variations. It contains several domains with different functions. Prediction of pathogenicity of amino acid substitutions with a reliable PON-P3 [31] indicated that about 70% are likely disease-causing. The longest region of benign deletions is located in the polyproline segment in the Tec homology (TH) region [64], which is likely intrinsically disordered [63]. This is indicated in the structure as a loosely packed string at the top of the protein structure (Fig 4B). Short deletions may not be harmful in this malleable region, which apparently binds to several partners, including the adjacent SH3 domain [65, 66]. The other benign deletions mainly occur in connecting loops or towards the ends of secondary structural elements. Experimental studies and structural information are in line with the deletion predictions. In addition, the PON-P3 predicted disease-causing substitutions are mainly pathogenic in the regions predicted not to tolerate the short deletions.

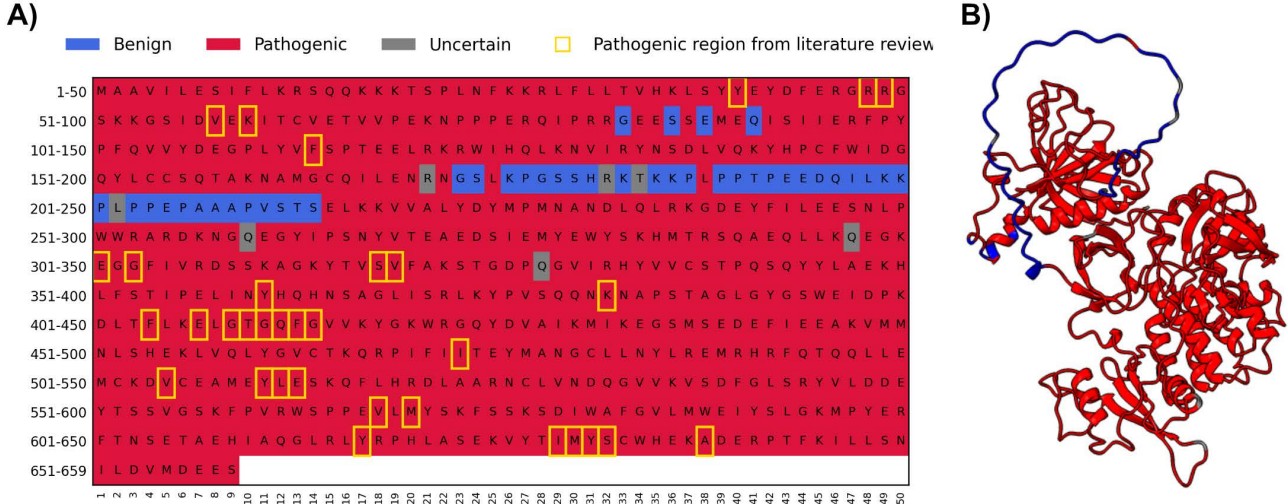

**Fig 4. A) Prediction of all the one-amino acid deletions in BTK.** Most variations are deleterious, except in the polyproline segment in the TH region. This region is disordered and can adopt numerous different conformations. Yellow boxes indicate the positions of the XLA-causing short deletions listed in BTKbase. **B)** Distribution of the pathogenic and benign variants in the BTK structure, obtained with AlphaFold2 [40], file AF-Q06187-F1-model_v4(1). The pleckstrin homology domain is located at the top left, below it are the Src homology 3 and 2 domains, and the kinase domain is positioned to the right. Known XLA-causing deletions are shown in yellow, predicted benign single amino acid deletions are in blue and predicted pathogenic deletions in red. α-Helices are shown as helices and β-strands as arrows. Benign variants are indicated in blue, pathogenic ones in red, and VUSs in grey.

## Availability and future directions

PON-Del is freely available as a web service at http://structure.bmc.lu.se/pon_del. Unlike most other deletion predictors, which accept only genomic coordinates, the variations can be submitted at the genomic, transcript, or protein level. The genome build used in PON-Del is GRCh38/hg38. Nucleotide variants are converted to protein alterations with TransVar. Only nucleotide deletions that lead to sequence retaining amino acid deletions are predicted. The allowed size range is 1–10 amino acids. Exon skipping variants are not allowed; the nucleotide deletion cannot be more than 30 bases. The dataset for short exon skipping variants was too small to support the development of a reliable predictor. It is possible to submit up to 1000 variants at a time.

PON-Del is entirely MANE-based. Therefore, the submitted variants must be mapped to MANE reference sequences. This is because many features are specific to a variation position and its context. If a submitted variant is in a protein for which all features cannot be obtained, a note is provided that prediction is not possible. This happens, e.g., when proteins are unique to humans and lack evolutionary details, or when no protein structure is available.

Users can submit multiple variations simultaneously across multiple genes or proteins. It is even possible to download a Fig that contains all the one-amino acid deletions in a protein. These data can be searched by gene name, RefSeq transcript or protein ID, or Ensembl gene, transcript, or protein ID. The precalculated data are available for 19354 unique MANE-compliant sequences.

PON-Del successfully predicts the disease relevance of short sequence retaining deletions. Once more data are available, it will be possible to expand to longer deletions. It will also facilitate further benchmarking. In addition, it will be interesting to investigate how well the current version of PON-Del can extrapolate from the short deletions to longer ones.

When verified VUSs are defined in deletions, they can be used both for training and testing of further versions. The future developments will be highly dependent on additional data and annotations.

## Supporting information

**S1 Fig. Distribution of the sizes of deletions obtained from the four databases.** Only variants 10 amino acids or shorter were used for method development due to the low number of longer deletions. Benign variants are in blue; pathogenic variants are in red. A) The deletion length distributions, B) the distribution of the deletion numbers per protein, and C) the length distribution of deletions in training and test datasets.
(TIFF)

**S2 Fig. Hyperparameter optimisation of PON-Del.** (A) Performance metrics across different numbers of top-ranked features. Red lines indicate the median, and boxplots represent the variability across cross-validation folds. (B) Hyperparameter tuning using Optuna. Left: optimisation history showing the progression of AUC values over 100 trials. Right: relative importance of hyperparameters, indicating that the number of leaves, boosting type, and learning rate contributed most to model performance.
(TIFF)

**S3 Fig. The performance of PON-Del and other frameworks on five train-test splits of the dataset.**
(TIFF)

**S4 Fig. The effect of p-value threshold in the prediction of VUSs.**
(TIFF)

**S5 Fig. The performance of PON-Del and other methods on the different datasets collected.**
(TIFF)

**S6 Fig. The performance of PON-Del and other methods on five train-test splits of the dataset.**
(TIFF)

## Author contributions

**Conceptualization:** Mauno Vihinen.

**Data curation:** Haoyang Zhang, Mauno Vihinen.

**Formal analysis:** Haoyang Zhang, Muhammad Kabir.

**Investigation:** Haoyang Zhang, Mauno Vihinen.

**Methodology:** Haoyang Zhang, Muhammad Kabir, Mauno Vihinen.

**Project administration:** Mauno Vihinen.

**Software:** Haoyang Zhang, Muhammad Kabir.

**Supervision:** Mauno Vihinen.

**Validation:** Haoyang Zhang, Mauno Vihinen.

**Visualization:** Mauno Vihinen.

**Writing – original draft:** Haoyang Zhang, Mauno Vihinen.

**Writing – review & editing:** Haoyang Zhang, Muhammad Kabir, Mauno Vihinen.

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
