## [Decision Letter · Decision Letter 0]

30 Nov 2025

PON-Del PREDICTOR FOR SEQUENCE RETAINING PROTEIN DELETIONS

PLOS Computational Biology

Dear Dr. Vihinen,

Thank you for submitting your manuscript to PLOS Computational Biology. After careful consideration, we feel that it has merit but does not fully meet PLOS Computational Biology's publication criteria as it currently stands. Therefore, we invite you to submit a revised version of the manuscript that addresses the points raised during the review process.

We look forward to receiving your revised manuscript.

Kind regards,

Mohammad Sadegh Taghizadeh, Ph.D.

Academic Editor

PLOS Computational Biology

Nir Ben-Tal

Section Editor

PLOS Computational Biology

**Journal Requirements:**

1) Your manuscript is missing the following sections: Design and Implementation, and Availability and Future Directions. Please ensure that your article adheres to the standard Software article layout and order of Abstract, Introduction, Design and Implementation, Results, and Availability and Future Directions. For details on what each section should contain, see our Software article guidelines:

https://journals.plos.org/ploscompbiol/s/submission-guidelines#loc-software-submissions

3) We notice that your supplementary Figure is included in the manuscript file. Please remove it and upload it with the file type 'Supporting Information'. Please ensure that each Supporting Information file has a legend listed in the manuscript after the references list.

4) Please provide a completed 'Competing Interests' statement, including any COIs declared by your co-authors. If you have no competing interests to declare, please state "The authors have declared that no competing interests exist". Otherwise please declare all competing interests beginning with the statement "I have read the journal's policy and the authors of this manuscript have the following competing interests:"

**Reviewers' comments:**

Reviewer's Responses to Questions

Reviewer #1: This paper examines diverse machine-learning models and feature sets to build a predictor capable of classifying short sequence-retaining protein deletions as benign, pathogenic or uncertain significance.

1. If the authors aim to answer the question of how a binary classifier can abstain from prediction when it is not confident, threshold-based selective prediction would be an easy and effective approach.

2. The authors extend a binary classifier into a three-state predictor by introducing a variant of uncertain significance (VUS) category, assigning VUS labels only after the model has been trained. A similar VUS assignment could be added to any competing tool; therefore, the VUS category is not unique to PON-Del.

3. In Table 4, the performance of the three-state classifier is no better than the binary version on any metric, including false positives and false negatives, which raises questions about the practical value of adding the VUS state.

Reviewer #2: In this paper, the authors develop a novel software to identify benign and pathogenic mutations from amino acid sequences ranging 1-10 residues. The developed algorithm relied on a deep learning framework trained and tested on a curated database of deletions. The particular framework to generate the prediction and the set of features were selected and optimised to improve the model’s predictive power. This tool was then compared to alternative software, displaying a modest improvement in some performance descriptors but not all. I found that the tool might be useful for the community, but the lack of clarity and the simplistic benchmarking are major issues on the current form of the manuscript.

General comments:

1) The dataset of variants used in this study after filtering consisted of 3912 deletions from 1904 proteins. This was further split into a training set with approx 80% of the data, while the other 20% was used for the blind test. The only specific criteria specified was that no protein was common between training and test sets and that the distribution of deletion lengths were similar. There is no further explanation on how the variants were assigned to each group. Besides, the lack of a cross-validation scheme significantly reduces the reliability of this model. It is not clear, for example, why a 5-fold CV was not used directly in the full dataset. If this was the case, then the proposed tool could be used multiple times to provide a more robust assessment of the tool and the concurrent methods with different training-test pairs.

2) While the authors explored which features were learned by the training, a similar evaluation was not performed to understand where the model fails. The outcome of the predictions (TP, TN, FP, FN) was not characterised. It would be important to observe if there are biases on the correctness of the prediction depending on protein families, database of origin, relative position of the mutation, etc. Also, does the (mis)classification pattern is correlated across tools?

3) One of the confusing and maybe misleading information I found is that, in the introduction, there is a detailed differentiation between sequence-retaining and amphigoric deletions. A range of examples on the possible effects of the latter, such as NMD and alternative splicing, were explained. Because the publicly available tool can also analyse transcript and genomic information, it seems that such phenomena can be detected by the software. However, after the introduction, only sequence-retaining deletions are mentioned throughout the text. Overall, it seems that the authors claim that such amphigoric deletions are neglected in most studies, but I do not see how it has been incorporated into this tool, or if it has at all.

4) The quality metrics (AUC, PPV, NPV, …) were applied to the data and are also displayed in brackets normalised by adjusting the number of benign and pathogenic variants. However, it has been said that most deletions are likely pathogenic. Therefore, this proposed normalisation is far from representing the conditions of a random sample of protein deletions. This can lead to spurious metrics with non-realistic quality assessments. I suggest that this normalisation is either removed or modified to accommodate criteria that better represents the known distribution of benign and pathogenic deletions.

5) It is not clear what is the impact of the filtering criteria to build the dataset over the assessment across different tools. These choices might negatively influence the performance of concurrent tools. The comparison seems unfair because PON-Del was trained and tested in data following the same filtering criteria, while the other tools did not.

6) The difference in performance between PON-Del 2-state and PON-Del 3-state is small, but sufficient to make PON-Del 3-state worse than at least one other method in each metric. Therefore, I suggest this differentiation is clear throughout the text and especially in the abstract.

Specific comments:

1) Is OPM a standard metric in the field? If so, it should be referenced. Otherwise, it could have been arbitrarily chosen in favor of PON-Del.

2) It would be interesting to see the application to a test case using PON-Del 3-state.

3) In figure 3B, it is not clear how the coloring is done. Maybe use four colors?

4) In table 2, the word metrics has a typo.

All the best.

Reviewer #3: The manuscript presents an interesting program addressing a complex challenge in biology: the predicted functional impact of variants on proteins. One of the main strengths of the work is that the authors evaluate several methods on a curated and cleaned dataset, allowing for a fair comparison of prediction algorithms in a specific scenario: namely, the effect of deletions of 1 to 10 amino acids on protein function. However, the main weakness of the study is precisely that this scenario is highly specific included in scenarios of other methods that operate beyond this narrow context and are designed to handle a wider spectrum of variants, including single-nucleotide variants (SNVs), frameshift (inducing deletions and insertions), and other types of indels. In contrast, the proposed method is built around a very constrained situation, i.e. deletions corresponding to amino-acid deletions leading to in-frame amino-acid loss. Interestingly, focusing on this restricted deletion context ultimately shows that some widely used methods already perform as well as possible when constrained to the conditions defined in this study, while still retaining the ability to operate beyond such a narrow context.

General comments on structure and clarity

The structure of the article raises issues of relevance and readability when compared with similar literature.

- The Introduction section contains methodological and concluding elements, which should be moved to the appropriate sections to improve narrative flow.

- The Results and Discussion sections are merged, making it difficult to clearly distinguish factual contributions from interpretative aspects. Splitting these sections would make the strengths, weaknesses, and contextual relevance of the method much clearer.

Comments on the core method

The proposed method performs at a level comparable to two reference approaches (FATHMM-indel and VEST-Indel). The choice of the PON-Del algorithm appears reasonable, although it does not fundamentally differentiate itself from other prediction methods comparing their validation features (Table 3). This lack of distinction is not explicitly discussed.

Among the available methods, several different algorithmic approaches are represented. Within the gradient boosting category, PON-Del appears to perform better than the other gradient boosting method (aka INDELpred). While other widely used tools such as FATHMM-indel, VEST-Indel, and SHINE achieve similarly strong performance despite relying on different algorithmic frameworks (hidden Markov models, neural networks, and transfer learning, respectively). This should also be presented in a discussion section, apart from the result section.

On the method validation: all method comparisons rely on a single evaluation using the blind set; however, increasing the size of this blind set and performing a bootstrap analysis across multiple resamplings of the blind set would provide valuable insight into the variability and robustness of performance differences between methods.

Another important point concerns the type of variants used for comparison. The reference methods process nucleotide deletions, which can introduce frameshifts, an aspect explicitly excluded for by PON-Del application. The manuscript rightly focuses on deletions of 1–10 amino acids, but this context-specific applicability must be clearly stated in a dedicated discussion section. The method appears optimal only for this niche use case, and this limitation should be openly discussed, especially since, in routine practice, deletions are not predominantly in-frame. While this does not diminish the technical value of the work, it raises questions regarding its broader applicability.

Finally, the training dataset should be more thoroughly and fairly discussed, including both its strengths and weaknesses. Although the methods section describes how the data were curated, it does not address which genes were included, whether certain genes are overrepresented, the biological processes in which these genes are involved, or the broader biological knowledge that the community has already established about them. This description would strengthen the trust in the training dataset (as well as in the blind set).

Online tool

The online implementation is user-friendly. However, it does not specify which genome version the tool accepts as input regarding the Genomic Position window. Concerning the output, as noted in the introduction, the handling of protein isoforms is not included. Since the position of an amino-acid deletion depends on the chosen isoform, it is necessary to explain whether and how isoform variability affects the algorithm’s prediction. Or include the multiple algorithm output that refers to isoforms.

The code is available on GitHub (python)

Specific remarks and questions

- “The methods displayed a wide range of performances. The MCC for the best methods was 0.68, far from perfect.” It is unclear whether this is the authors’ own analysis or derived from a referenced study.

- Inclusion of VUS is appreciated. However: Can a VUS become pathogenic depending on biological context? How could the algorithm handle this ambiguity?

- Several sentences in the introduction belong to the Methods, Conclusion, or Discussion sections.

- Figure 1: The different distribution of pathogenic vs. benign variant lengths suggests a potential bias in the original dataset. Or should be discussed.

- Table 3: MetRics (instead of metics); in addition, a bootstrap-based 95% confidence interval would improve the robustness of comparisons.

- Figure 3B: The red/blue legend is missing. Is the coloring original annotation or the one generated by PON-Del? If the latter, what about VUS classifications?

- Table 4: Again, confidence intervals (e.g., bootstrapped 95% CI) would strengthen the analysis.

**Have the authors made all data and (if applicable) computational code underlying the findings in their manuscript fully available?**

Reviewer #1: Yes

Reviewer #2: Yes

Reviewer #3: Yes

PLOS authors have the option to publish the peer review history of their article (what does this mean? ). If published, this will include your full peer review and any attached files.

**Do you want your identity to be public for this peer review?** For information about this choice, including consent withdrawal, please see our Privacy Policy .

Reviewer #1: No

Reviewer #2: No

Reviewer #3: **Yes:** Tosato G.

**Figure resubmission:**
---

## [Decision Letter · Decision Letter 1]

30 Jan 2026

PON-Del PREDICTOR FOR SEQUENCE RETAINING PROTEIN DELETIONS

PLOS Computational Biology

Dear Dr. Vihinen,

Thank you for submitting your manuscript to PLOS Computational Biology. After careful consideration, we feel that it has merit but does not fully meet PLOS Computational Biology's publication criteria as it currently stands. Therefore, we invite you to submit a revised version of the manuscript that addresses the points raised during the review process.

We look forward to receiving your revised manuscript.

Kind regards,

Nir Ben-Tal

Section Editor

PLOS Computational Biology

**Additional Editor Comments:**

The reviewers pointed out quite a few outstanding issues. It is surprising for a revised draft, and would normally deal to rejection. However, I'm giving you a chance to revise again and resubmit.

**Reviewers' comments:**

Reviewer's Responses to Questions

**Comments to the Authors:**

Reviewer #1: Thank you to the authors for their response and revisions. While some of my concerns were not directly answered. I outline them below for further clarification.

1. My primary concern remains whether the proposed method introduces unnecessary complexity for a problem where simpler solutions may already exist. In particular, a threshold-based selective prediction approach should be included as a baseline for benchmarking.

2. The manuscript reports a large number of evaluation metrics, which makes it challenging to understand the overall performance. I recommend that the authors separate comprehensive metrics (e.g., MCC) from metrics that reflect only specific aspects of performance.

3. In addition, my previous concern regarding the three-state classifier has not been adequately addressed. Based on the results presented in Table 4, the performance of the three-state classifier does not appear to be superior to that of the two-state classifier. The authors should therefore revise their claims to more accurately reflect the empirical evidence.

4. Finally, I note an inconsistency in the use of terminology: both PON-Del and PON-DEL appear in the manuscript. The authors should ensure consistent naming.

Reviewer #2: The authors have adequately addressed my previous comments, except for the major revision proposed in comment #4, which I am further clarifying below:

4) The quality metrics (AUC, PPV, NPV, …) were applied to the data and are also displayed in brackets normalised by adjusting the number of benign and pathogenic variants. However, it has been said that most deletions are likely pathogenic. Therefore, this proposed normalisation is far from representing the conditions of a random sample of protein deletions. This can lead to spurious metrics with non-realistic quality assessments. I suggest that this normalisation is either removed or modified to accommodate criteria that better represents the known distribution of benign and pathogenic deletions.

RESPONSE: When reporting the method performances, we followed published and widely followed guidelines for assessment (PMID: 22759650 and 3169447). Indeed, the distribution of most variation types is biased. Amino acid substitutions are mainly benign or VUS, many deletions are harmful, etc. A detailed study has shown that even in such cases, training and testing with a balanced dataset provides the best performance and insight (PMID: 23874456). If results are not normalised, when using an unbalanced dataset, some of the measures are severely affected, see PMID: 22759650.

Indeed, as well noted by the authors when citing reference PMID: 22759650, the normalisation often provides a better estimation of descriptive statistics when there is significant class imbalance. However, as written in the same paper, “ When normalizing the data be sure that the existing dataset is representative otherwise bias in the set may further be increased.” In this case, and repeating my original comment, the normalisation used by the authors (50% benign and 50% pathogenic) is far from reality. In other words, when a user applies this method to their own data, these estimates are not representative of the expected statistics. Therefore, I reiterate that the normalisation used should match the fraction of benign and pathogenic variants observed in reality.

In addition, I have two minor comments on the new content included in the newest version:

1) In the third paragraph of results, the authors say: The length differences were very small. But small relative to what? Is there a statistical difference? Besides, the difference between the mean length for training (4.11 - 2.71 = 1.4) and test (4.51 - 2.51 = 2) sets increased by 43% (2/1.4). Is this difference very small?

2) It is not clear to me, neither on the image nor in the legend, of Fig3 why the proportions of correct and misclassified predictions do not add to 100%.

Reviewer #3: I warmfully thank the authors to have patiently answered every point I mentioned so far and to have completed the main text when they have considered it appropriate. The addition of a bootstrap resampling approach strengthens the trust in the results and the cross-validation. The text structure makes the paper clearer.

The only addition would be the CI95% or any other variance representation for each performance feature in table 4. However Fig Supp 3 and 6 play that role and will be of interest to the thoughtfull readers.

Minor comment:

"Performance evaluation" title is misplaced in the previous section.

**Have the authors made all data and (if applicable) computational code underlying the findings in their manuscript fully available?**

Reviewer #1: None

Reviewer #2: Yes

Reviewer #3: Yes

PLOS authors have the option to publish the peer review history of their article (what does this mean? ). If published, this will include your full peer review and any attached files.

**Do you want your identity to be public for this peer review?** For information about this choice, including consent withdrawal, please see our Privacy Policy .

Reviewer #1: No

Reviewer #2: No

Reviewer #3: **Yes:** Guillaume Tosato

**Figure resubmission:**

**Reproducibility:**



---

## [Decision Letter · Decision Letter 2]

13 Feb 2026

Dear Prof. Vihinen,

We are pleased to inform you that your manuscript 'PON-Del PREDICTOR FOR SEQUENCE RETAINING PROTEIN DELETIONS' has been provisionally accepted for publication in PLOS Computational Biology.

Best regards,

Nir Ben-Tal

Section Editor

PLOS Computational Biology

Nir Ben-Tal

Section Editor

PLOS Computational Biology

Reviewer's Responses to Questions

**Comments to the Authors:**

Reviewer #2: I thank the authors for clarifying the remaining doubts and for modifying the text with the appropriate changes.

**Have the authors made all data and (if applicable) computational code underlying the findings in their manuscript fully available?**

Reviewer #2: Yes

PLOS authors have the option to publish the peer review history of their article (what does this mean? ). If published, this will include your full peer review and any attached files.

**Do you want your identity to be public for this peer review?** For information about this choice, including consent withdrawal, please see our Privacy Policy .

Reviewer #2: No

---

## [Editor Report · Acceptance letter]

PCOMPBIOL-D-25-02232R2

PON-Del PREDICTOR FOR SEQUENCE RETAINING PROTEIN DELETIONS

Dear Dr Vihinen,

I am pleased to inform you that your manuscript has been formally accepted for publication in PLOS Computational Biology. Your manuscript is now with our production department and you will be notified of the publication date in due course.

With kind regards,

Anita Estes
